# Real-Life Wheelchair Mobility Metrics from IMUs

**DOI:** 10.3390/s23167174

**Published:** 2023-08-14

**Authors:** Wiebe H. K. de Vries, Rienk M. A. van der Slikke, Marit P. van Dijk, Ursina Arnet

**Affiliations:** 1Swiss Paraplegic Research, Guido A. Zächstrasse 4, 6207 Nottwil, Switzerland; ursina.arnet@paraplegie.ch; 2Department of Biomechanical Engineering, Delft University of Technology, 2628 Delft, The Netherlands; r.m.a.vanderslikke@hhs.nl (R.M.A.v.d.S.); m.p.vandijk@tudelft.nl (M.P.v.D.); 3Human Kinetic Technology, The Hague University of Applied Sciences, 2521 The Hague, The Netherlands

**Keywords:** IMU, wearable sensors, spinal cord injury, activities of daily living, wheelchair propulsion, shoulder

## Abstract

Daily wheelchair ambulation is seen as a risk factor for shoulder problems, which are prevalent in manual wheelchair users. To examine the long-term effect of shoulder load from daily wheelchair ambulation on shoulder problems, quantification is required in real-life settings. In this study, we describe and validate a comprehensive and unobtrusive methodology to derive clinically relevant wheelchair mobility metrics (WCMMs) from inertial measurement systems (IMUs) placed on the wheelchair frame and wheel in real-life settings. The set of WCMMs includes distance covered by the wheelchair, linear velocity of the wheelchair, number and duration of pushes, number and magnitude of turns and inclination of the wheelchair when on a slope. Data are collected from ten able-bodied participants, trained in wheelchair-related activities, who followed a 40 min course over the campus. The IMU-derived WCMMs are validated against accepted reference methods such as Smartwheel and video analysis. Intraclass correlation (ICC) is applied to test the reliability of the IMU method. IMU-derived push duration appeared to be less comparable with Smartwheel estimates, as it measures the effect of all energy applied to the wheelchair (including thorax and upper extremity movements), whereas the Smartwheel only measures forces and torques applied by the hand at the rim. All other WCMMs can be reliably estimated from real-life IMU data, with small errors and high ICCs, which opens the way to further examine real-life behavior in wheelchair ambulation with respect to shoulder loading. Moreover, WCMMs can be applied to other applications, including health tracking for individual interest or in therapy settings.

## 1. Introduction

Manual wheelchair users often report persistent shoulder pain and problems. The prevalence of shoulder pain is accordingly high, ranging from 39 to 71% [1,2,3]. Not only is the pain itself a relevant problem. Since manual wheelchair users are dependent on their upper extremities for many aspects of daily life, shoulder health is also highly correlated with independence, social participation and quality of life [4,5,6].

Wheelchair users are 5.8 times more likely to suffer from shoulder pain than able-bodied controls [1]. Due to the inability to use the lower extremities, wheelchairs users may put increased load and repetitive stress on their shoulders during many activities of daily life (ADL), such as during wheelchair propulsion [7], pressure relief lifts [8] and transfers [9]. The shoulder is a very mobile joint, not built for the high and frequent loads experienced in the daily life of a wheelchair user. If the demands of daily life exceed the capacity of the shoulder, it will lead to acute or overuse injuries. Both are very frequent in this population [10].

Wheelchair propulsion as a risk factor for shoulder problems has been the focus of previous investigations. Several studies have been conducted to address the amount of shoulder load in wheelchair propulsion [8,11,12] or the relationship between wheelchair propulsion characteristics and shoulder pathologies [13,14] or shoulder pain [1,14,15,16]. Usually, these studies evaluate wheelchair propulsion in a cross-sectional design during a standardized protocol in a laboratory setting, which provides a very detailed understanding of the amount of shoulder load during this specific task. The amount of shoulder load is one of the three factors needed to evaluate the risk for shoulder problems, along with the frequency and duration of the exposure. It is essential that these factors are assessed over a longer period (days or weeks) in the daily lives of the wheelchair users where shoulder problems are emerging.

IMUs are ideal for monitoring wheelchair propulsion unobtrusively over a long period of time. In comparison with measurement wheels, such as the Smartwheel, which have been used to quantify wheelchair propulsion in a laboratory setting [17,18,19], IMUs have the advantages of being lightweight, easily attached wheelchairs, able to measure over a longer time and less costly. IMUs measure three-dimensional acceleration and angular velocity. Therefore, the disadvantage of IMUs is that propulsion variables are not measured directly on the wheel but must be deduced from the (raw) IMU signals. To use IMUs for tracking physical activity and wheelchair propulsion, methodologies have been developed for use in both daily life [20,21,22,23,24,25,26,27,28,29,30,31,32,33,34,35,36] and sports settings [37,38,39,40,41,42]. Regarding wheelchair propulsion in daily life, most of the literature is published on methods for quantifying a manual wheelchair user’s energy expenditure or activity levels during physical activity [27,30,31,34]. However, methods for quantifying wheelchair propulsion itself, and thus quantifying the wheelchair user’s exposure to one of the risk factors for shoulder problems, are scarce. Nevertheless, in general, one or two variables of propulsion quantification are addressed in previous publications, (such as speed [26,29,36], distance covered [26,29,33,35,36], number of pushes [20,24,25,28,32], duration of movement [23,36], direction of movement [23] or propulsion style [21,22]); a methodology combining several of the relevant variables is still missing.

In addition to quantifying wheelchair propulsion for studying the risk of shoulder overload in research, wheelchair users also wish to use such technologies for self-care. Unlike other chronic health conditions, health-tracking solutions for shoulder health management are not yet commercially available [16]. Li et al. showed that such health-tracking technologies are desired and can support manual wheelchair users’ self-care. The accessibility of tracking devices or applications can provide motion awareness, support the understanding of one’s own capacity and can facilitate communication with therapists. The knowledge of wheelchair propulsion data and their trends can further inform activity planning to prevent overuse injury [16] or analyze the outcome of health-related interventions in the domain of activity and participation [43,44]. The tracking features desired by wheelchair users and therapists are the measurements of push speed, distance and duration, stroke efficiency and propulsion patterns [16]. Preferably, tracking devices will be unobtrusive, easy to handle and not interfere with performing the activities of daily life.

Summing up, the methods for quantifying wheelchair propulsion in daily life are still scarce and usually only address some of the features desired by wheelchair users and therapists or some of the features needed for evaluating wheelchair use as a risk factor for shoulder problems. In this study, we aim to describe and validate a comprehensive and unobtrusive methodology to derive a full set of wheelchair mobility metrics (WCMMs) from one or two IMUs placed on the wheelchair while collecting data in a real-life setting. The described WCMMs include distance covered by the wheelchair, linear velocity of the wheelchair, number and duration of pushes, number and magnitude of turns and inclination of the wheelchair when on a slope.

## 2. Materials and Methods

### 2.1. Study Sample

Ten able-bodied participants (3 male, 7 female, age 39 ± 9.4 years, height 169 ± 9.1 cm, weight 66 ± 12.0 kg) were invited to participate in the experiment. The participants were first trained in the wheelchair-related tasks of interest for several hours. Such an able-bodied study sample cannot be representative of the manual wheelchair user population in all of its aspects, for instance considering trunk stability, which in the SCI case might be affected by the lesion level but is still available in the able-bodied case. However, as indicated by Vegter et al. [45], several aspects, such as wheelchair propulsion characteristics, do converge to patterns observed in manual wheelchair users within the first 12 min of training in novice wheelchair users. The study was approved by Ethikkommission Nordwest- und Zentralschweiz (EKNZ, Project-ID: 2020-01961). Informed consent was obtained from all subjects involved in the study.

### 2.2. Data Collection

A standard active wheelchair (Küschall K-Series 2005, Küschall AG, Witterswil, Switzerland) was equipped on the right side with a 24-inch Smartwheel (SW, Three Rivers Holdings, Inc., Mesa, AZ, USA; measuring wheel rotation and the forces and torques exerted at the rim at 240 Hz) and 2 Shimmer IMUs (IMU, Shimmer3 IMU unit, Shimmer, Dublin, Ireland; measuring acceleration and angular velocity at 100 Hz) on the wheelchair frame and the right wheel, as depicted in Figure 1. The Shimmer IMUs were configured as follows: wide-range accelerometer range 8 g at 16-bit resolution, having an RMS noise value of 27.5 × 10^−3^ m/s^2^. The gyroscope range was set at 2000 degrees/s, at 16-bit resolution, having an RMS noise value of 0.0481 degrees/s.

While following a course around the campus, the participants implemented a variety of speeds, turns and slopes and completed a figure-8 maneuver 3 times. All trials were recorded on a consumer-grade video recorder for the referencing of activities. Data were synchronized by a specific procedure: “the mechanical IMU sync”. For that procedure, all the IMUs were attached to a rigid beam that was rotated around the longitudinal axis 10 times before and after the actual measurements. This delivers a clear recognizable pattern in the gyroscope signals that can be synchronized by cross-correlation. Visual inspection of the peaks of the synchronized signals indicated an accuracy of one to two samples (which equals 10–20 ms). While using simple and straightforward signal processing and calculations, the next seven WCMMs were derived from the acquired IMU data for comparison with reference methods:Distance covered;Linear velocity of the wheelchair;Number of pushes;Duration of pushes;Number of turns;Magnitude of turns;Wheelchair inclination.

The IMU-derived WCMMs were validated against appropriate reference methods (Smartwheel data, video analysis, inclinometer) as described below. All data were collected in real-life conditions.

### 2.3. Preprocessing of IMU Data

We aimed for a minimum of preprocessing for the IMU data; however, besides basic filtering of noise (2nd order Butterworth bidirectional lowpass filter with a cut-off frequency of 5 Hz) [46], two corrections were required for proper calculation of most of the variables: the alignment of the frame IMU with respect to gravity and the determination of the alignment of the wheel IMU with respect to the wheel axis.

The IMU mounted on the frame of the wheelchair was manually aligned with one of the horizontal lateral pointing tubes of the frame; however, vertical alignment with gravity proved to be hard. For that reason, a period of 10 s of complete standstill (on level floor) was selected from the recorded data (both frame and wheel). Averaging the measured acceleration over these 10 s delivered a steady measure of gravity, which was used to align the IMU’s frame of reference with gravity.

Similarly, the IMU mounted on the wheel could not be accurately aligned with the wheel axis. Therefore, a period of 10 s of straightforward wheelchair ambulation was selected from the recorded data for every participant. This track of straightforward ambulation was selected from the video recordings and was the same trajectory for all participants. Averaging the measured angular velocity over these 10 s delivered a steady measure of the rotation around (and a definition of the vector of) the wheel axis in the IMU frame of reference. Concurrently, the angular velocity around the wheel axis can be calculated conform Equation (1).
IMU_AV_axis = cos (α) × IMU_AV_tot(1)
where:

IMU_AV_axis = angular velocity around the wheel axis from wheel IMU;

α = angle between vector of the wheel axis in IMU reference system and the currently measured angular velocity vector;

IMU_AV_tot = total angular velocity as 3D-vector sum.

### 2.4. Derivation of WCMM Variables from Reference Methods and IMU Data

#### 2.4.1. Distance Covered

The Smartwheel contains an optical rotary encoder measuring wheel angle (SW_A), accumulated over time. This variable was reset to zero at the start of measurement. Concurrently, the distance covered can be calculated following Equation (2).
SW_Dis = (SW_A/360°) × WD × π(2)
where:

SW_Dis = distance covered according to Smartwheel;

SW_A = Smartwheel angle accumulated;

WD = Wheel diameter in meters.

From the wheel IMU data, integration over time of the angular velocity around the wheel axis delivers the accumulated wheel angle over time, from which the distance covered can be calculated in the same fashion as for the Smartwheel, following (Equation (2)). ICC was applied to test the reliability of the estimation of the distance covered for both systems across participants (two-way mixed, single measures, participants as random factor).

#### 2.4.2. Linear Velocity

The angular velocity can be obtained from the Smartwheel by differentiating the Smartwheel angle accumulated over time. For both the Smartwheel and IMU data, angular velocity around the wheel axis (expressed in °/s) can be used to calculate the linear velocity in the same fashion as distance covered (Equation (2)), by replacing accumulated angle with angular velocity. The linear velocities from both systems were compared by calculating the RMS value over the complete signals.

#### 2.4.3. Number and Duration of Pushes

There are several methods for push detection from Smartwheel data, ranging from simply applying a threshold above noise level [2] at the propulsive torque signal to a more elaborated four-phase push detection defining hand contact from the measured force and discerning between braking and propulsive force [3,4]. For the sake of simplicity in comparing the two systems used (Smartwheel and IMUs), the basic approach for push detection was followed for the Smartwheel data. We applied a threshold of 2 Nm on the propulsive torque measured for positive rolling velocities.

For push detection from IMU data, the following basic assumption was made: in all-day wheelchair propulsion, when propulsive force is exerted at the rim, the angular velocity will increase. When there is no longer a force exerted, the angular velocity will decrease due to resistive forces. A basic peak detection algorithm (standard Matlab routine from the Signal Toolbox, R2021a, Mathworks, Natick, MA, USA) was used to first detect the valleys in the signal of angular velocity around the wheel axis (AV-axis) and concurrently the peaks in between the valleys. This was applied for forward propulsion at those time instances where the angular velocity was positive. The difference between the start and end of each push determined the push duration. The same assumption was the basis for push detection from the angular acceleration (AA), where a push-start is detected when the AA is positive and a push-end is detected when the AA is negative.

For both systems (Smartwheel and IMU data), a push was counted as a push when it resulted in a more than 30° rotation of the wheel. Furthermore, pushes with a push duration of more than 2 s were excluded.

ICC was applied to test the reliability of the estimations of the number of pushes and the average push duration across participants (two-way mixed, single or average measures where applicable, participants as random factor).

#### 2.4.4. Number and Magnitude of Turns

All participants performed a figure-8 maneuver like those in Bossuyt et al. [5] and Collinger et al. [6]; however, this maneuver was only completed three times as fatigue induction was not aimed for. Since the Smartwheel cannot measure the amount of turn around the global vertical (WC_turn), another reference method was applied here. Using the video recordings of the figure-8 maneuver and a 2D-DLT method [7], the 2D orientation of the wheelchair was calculated at the start and end of the figure-8 maneuver and compared with the IMU data.

From the wheelchair frame IMU (after alignment with gravity), the angular velocity around the vertical was time integrated and the continuous angle of turn (either positive or negative depending on left or right turns) accumulated from the start to end of the protocol. The resulting WC_turn from the wheelchair frame IMU was compared with the difference in 2D orientation of the wheelchair obtained from the 2D-DLT method. WC_turn was compared between systems by applying ICC (two-way mixed, single measures, participants as random factor).

The continuous angle of turn from the wheelchair frame IMU served as a reference for the number and magnitude of the WC_turn for the whole of the course measured. WC_turn can be derived from the wheel IMU’s angular velocity signal in a simple fashion based on the following two assumptions: (1) daily wheelchair use does not include sidewards tipping of the wheelchair; (2) the number and inclinations of sidewards-oriented slopes are ignorably small compared with the total amount of vertical turn a wheelchair user encounters on a daily base. Any deviation from the actual measured angular velocity from the wheel axis can then be attributed to a rotation around the global vertical (a WC_turn) and can be calculated using Equation (3). However, the direction of turn (left or right) is lost in this way. Time integration of the signal delivers a continuous angle of turn around the global vertical (but absolute values only).
IMU_AV_turn = sin (α) × IMU_AV_tot(3)
where:

IMU_AV_turn = angular velocity of wheelchair turning around the global vertical;

α = angle between the vector of the wheel axis in IMU reference system and the currently measured angular velocity vector;

IMU_AV_tot = total angular velocity as 3D-vector sum.

From the continuous angle of turn, the number of turns and the magnitude thereof can be extracted for both IMUs (frame and wheel). To enable comparison of the magnitude, the continuous angle of turn derived from the frame IMU was rectified. Only turns larger than 30° were included. The absolute number of turns was compared with the ICC between the frame- and wheel-IMU-derived values across participants (two-way mixed, average measures, participants as random factor).

Comparison of the calculation of the magnitude of WC_turn between the frame and wheel IMUs was performed by calculating the intersection of the two normalized histograms of the turn [8] according to Equation (4).
(4)$$I\left(y,\;\hat{y}\right)=\frac{\sum_{i=1}^{n}\min\left(y_{i},\;{\hat{y}}_{i}\right)}{\sum_{i=1}^{n}y_{i}}$$

#### 2.4.5. Wheelchair Inclination

The different inclines of the course were measured with a digital inclinometer (SPI-Tronic Pro 360) at 5 equidistant locations, for which the values were averaged. As the frame IMU was aligned with gravity, the angle between the actual acceleration measured and the vertical axis of the IMU contains information on incline besides horizontal acceleration due to wheelchair propulsion. To separate these two types of information, a 4th-order bidirectional low-pass Butterworth filter with a cut-off frequency of 0.2 Hz was applied to the frame IMU’s acceleration signal. This cut-off frequency was determined by experimentation and considering the average time needed to propel over the inclination. The inclination was calculated from IMU data as the angle between the filtered acceleration signal and the IMU’s vertical axis in the sagittal plane, averaged over the course of the slope. The calculated inclinations from the IMU data were compared with the (digital level) measured inclination by ICC (two-way mixed, average measures).

## 3. Results

### 3.1. Study Sample

In total, 302 min of wheelchair activities by the 10 participants was analyzed. The summed distance covered on campus was around 6.5 km. The overall qualitative performance of the participants was highly variable. Some participants, even after several hours of wheelchair training, were still careful and slow in propelling their wheelchair on flat surfaces, on slopes and in making turns, whereas other participants were clearly more experienced and showed much more dynamic behavior. This variation in performance is comparable with the variation in propulsion patterns in the target population of wheelchair users when including persons with paraplegia and tetraplegia. As mentioned before, the able-bodied study sample cannot be seen as representative of the manual wheelchair user population in all of its aspects, for instance considering trunk stability. However, the exerted forces at the rim from the current study sample followed very similar patterns to those observed in data obtained from studies with persons with SCI. Furthermore, in the current study, the movements of the wheelchair are measured and compared using two different devices and methodologies. Despite the variation in, for instance, the propulsion patterns observed in the study sample, the two methods showed good comparability.

### 3.2. Derivation of WCMM Variables from Smartwheel and IMU Data

The results of the comparisons of the derived WCMM variables, according to error measures and ICC (where applicable) are summarized in Table 1. The reference and IMU values are depicted as group averages in the table; however, error measures were derived from, and ICC were applied to, individual values across participants. Most WCMMs show good agreement, except for push detection from wheel IMU angular acceleration and push duration detection from both angular velocity and acceleration of the wheel IMU.

#### 3.2.1. Distance Covered

The total distance covered by both the Smartwheel and wheel IMU appeared to be close to equal and cumulated equally over time, as indicated for one of the participants by Figure 2.

#### 3.2.2. Linear Velocity

In a similar fashion, the linear velocity of the wheelchair, as derived from the Smartwheel data, appeared to be close to equal to the linear velocity derived from the wheel IMU data, with an average RMS value of 0.02; a typical example is depicted in Figure 3.

#### 3.2.3. Number and Duration of Pushes

Push detection was performed on propulsive (positive) torque from the Smartwheel at positive velocities only, as most studies examine wheelchair propulsion characteristics on forward ambulation. Similarly, push detection from the IMU data was defined for increasing, positive velocities. Push detection from IMU_AV_axis appeared to be the most robust in comparison with the basic threshold-based algorithm on propulsive torque for Smartwheel data, with an average MAND of 4% (Table 1). A typical example is depicted in Figure 4.

Push detection from the IMU_AA_axis signal resulted in around a 20% higher number of pushes than from the Smartwheel data, although it delivered a shorter push duration than from the IMU_AV_axis signal (Table 1). However, the push durations derived from the IMU data were longer than those from the Smartwheel data: about 60% for AV and around 30% for AA-derived push duration. Consequently, the overlap of the normalized histograms comparing the Smartwheel- and IMU-derived push durations showed lower values of 51% and 60% for the AV and AA signals, respectively, as depicted in Figure 5.

#### 3.2.4. Number and Magnitude of Turns

As the total turn from the figure-8 maneuver from the frame IMU data differed by less than 1% from the 2D-DLT derived total turn, it seemed appropriate to use the frame IMU as a reference for the wheel-IMU-derived amount of turn. The overlap of the normalized histograms on magnitude of turns was, on average, 95.9%; this is shown for one participant in Figure 6.

#### 3.2.5. Wheelchair Inclination

Wheelchair inclination on longer slopes was detected with a small error (0.3°) and high reliability (ICC = 0.975).

## 4. Discussion

### 4.1. Study Sample

Despite the high variation in performance of participants, the derivation of most WCMM variables was robust and consistent in comparison with the values from the respective reference systems; all data were collected in real-life conditions. This is a promising result, enabling the future application of the described methodology for collecting WCMMs from IMU data in real-life settings.

### 4.2. Derivation of WCMM Variables from the Smartwheel and IMU Data

#### 4.2.1. Distance Covered

The distance covered shows a low error of −0.2% between methods and a high reliability, with ICC of 1.00. Therefore, the distance covered derived from an IMU on the wheel of the wheelchair can be considered equal to the distance derived from a Smartwheel. This variable can be used as a primary measure of mobility assessment, a simple but robust variable for life space assessment, as described by Giannouli et al. [47]. Although only a one-sided measurement was performed, this gives a good estimate of distance traveled. Such a measurement is not the same as the distance covered for the center of the wheelchair when making turns, which happens frequently in real-life settings. When an accurate measurement of the distance covered by the wheelchair and its user is required, it is usually defined as the distance that the center of the wheelchair has traveled and a method with two IMUs should be applied [42]. The daily distance covered can be used as a rudimentary measure for wheelchair mobility, in the same fashion as a step counter for the general population.

#### 4.2.2. Linear Velocity

The linear velocity data from the IMU on the wheel had a low RMS of 0.02 when compared with the values from the reference system. This variable can then be further detailed into bouts of mobility or activity and rest times. Since this method measures continuously, it allows for flexibility in the selection of bout durations for research or treatment purposes.

#### 4.2.3. Number and Duration of Pushes

The number of pushes detected from the wheel IMU’s angular velocity differs by about 4% in comparison with the propulsive-torque-based push detection from the Smartwheel. With an ICC of 0.989, the methods can be considered equal. The number of pushes detected from angular acceleration is, however, around 20% higher than from the Smartwheel data and is not recommended. Furthermore, the underlying assumption or application area for the described method is focused on counting pushes during active wheelchair propulsion; being pushed is not discerned and probably needs more complex methods [33]. The number of pushes a day is an interesting measure in relation to general activity (such as distance covered); however, since each push is delivered by the upper extremity, it is also an interesting variable in relation to eventual shoulder problems, as a first indication of the daily shoulder load. As such, the monitoring of daily wheelchair behavior in terms of the WCMMs described in this publication can enrich the future research on shoulder problems in MWUs.

The push duration values derived from the IMU data are, however, different from the push duration values derived from the Smartwheel; this can be observed in Figure 5, where the histograms shift to “longer duration”. This can be explained as follows: the Smartwheel measures the torque exerted by the hands at the rim, whereas an IMU measures the total wheelchair acceleration. The latter is also influenced by trunk and arm movement, potentially leading to increased velocity/acceleration during the recovery phase [48] and thereby lengthening the acceleration phase. The difference in push duration derived from the two systems (Smartwheel and IMU) has consequences for the calculation of combined variables such as power output, for which push duration is a required variable when expressed per push. Furthermore, the rolling resistance should be measured and incorporated for proper calculation of power output. Time normalized profiles of applied force at the rim (or an estimate thereof) depend on push duration detection to segment and normalize the measured data. Such force profiles can give additional information on wheelchair propulsion technique, as rate of rise or jerk are shown to be associated with shoulder pain [15]. When the goal of data collection is purely focused on shoulder load, for now it is preferable to use a Smartwheel for force exertion (or propulsive torque) measurements. Future work might lead to the development of more advanced algorithms for the detection of propulsive torque from IMU data.

As no clear definitions of a wheelchair push in real-life settings could be found in the literature, we defined a threshold of a minimum of 30° of wheel rotation per push detected, as well as a cut-off of a maximal push duration of 2 s. These thresholds and cut-offs were applied to data from both the Smartwheel and IMUs but are in fact “food for discussion”: what instance should be considered as a push in wheelchair propulsion in daily conditions, in terms of minimal wheel rotation, increase in angular velocity, and/or duration of the push? The cut-off of a maximum 2 s for push duration was based on visual inspection of the data and the corresponding video for the longer pushes detected, which appeared to be caused by acceleration of the wheelchair when going downhill. As the algorithm used defines the end of a push when the angular velocity starts decreasing, the end of the push was, in these instances, detected when braking or reaching the end of the ramp or longer incline. The same discussion should be held regarding the definition of a turn in wheelchair ambulation, in terms of minimal amount of turn or duration. For instance, should a correction in driving direction of 5° count as a turn? It is expected that future monitoring of WCMMs will add to increased knowledge on a clinically significant amount of turn.

#### 4.2.4. Number and Magnitude of Turns

The turning of the wheelchair can be reliably calculated from IMU data, either from a IMU attached to the wheelchair frame or a wheel mounted IMU, under the assumptions described. WC turning is an immature area of research in manual wheelchair propulsion, where peak forces at the rim can double in comparison with straightforward propulsion [49]. Furthermore, Togni et al. identified 900 turns a day, on average, for a sample of 14 wheelchair users who were monitored for a week. Measuring WC_turn from real-life behavior can therefore further increase insight into the magnitude and frequency of shoulder load in MWUs. In daily wheelchair use, a turn is usually made by pushing harder at one side than the other. However, in wheelchair sports, a turn is more often initiated by braking on one side. Different types of turning strategies could also be observed in the real-life data collected in this study, based on a more extensive analysis of the push detection from the Smartwheel data.

In fact, six different push styles were observed in interaction with different turning styles: application of propulsive torque when driving forward, changing driving direction (from forward to backward or vice versa), which is usually the case when making a turn from standstill, or applying propulsive torque when driving backward, which in fact is braking of the wheelchair, or making a turn. The same can be observed for the application of braking torque for the three driving directions described above. Figure 7 depicts the above described for one participant. From this figure, it can be seen that 126 pushes (26 + 2 + 75 + 12 + 11) out of 439 (total) were not propulsive when driving forward. This stretches the need for clear definitions of pushing and braking, especially for the SCI population, as some persons with SCI lack trunk stability, which complicates braking actions. The inclusion of only able-bodied participants for this project had some limitations towards the type of pushing/braking observed in the data, as all subjects had trunk function. Future research should address the detection of these different types of pushing/braking actions from IMU data.

#### 4.2.5. Wheelchair Inclination

The inclination on longer slopes on the campus could be measured accurately, with an error of 0.3°. Only three participants agreed to descend a ramp with an 11° inclination, where two of them tipped the wheelchair backwards to control speed during the descent. For those situations, wheelchair inclination is still measured correctly but is no longer representative of the slope or ramp negotiated. Nevertheless, as propelling uphill usually occurs without tipping for speed control and puts increased loads at the shoulder, wheelchair inclination is an interesting variable to measure; however, this requires an extra IMU on the wheelchair frame.

### 4.3. Future Research

The variation in pushes and braking actions observed from the real-life data collected deserves further research and the development of appropriate calculations or algorithms to detect the variety of pushes. Braking actions or turning by braking can put higher loads on the shoulder, and proper execution of turns and training to do so could help in reducing shoulder problems. Furthermore, detection of the mounting/descending of roadside curbs or maneuvering over irregular surfaces or uneven terrain, as higher shoulder loading activities, could provide additional insight and help to optimize shoulder loading behavior from wheelchair ambulation. The monitoring of daily wheelchair use in terms of the described WCMMs could add valuable insight into daily shoulder load on modifiable variables. For instance, the quantification of the daily distance covered or the number of pushes a day deliver very clear variables on the mobility of MWUs. The number of wheelchair turns in daily conditions and magnitude of them, might shed more light on shoulder load from daily wheelchair ambulation as a potential undervalued risk for shoulder problems. The amount and steepness of inclinations encountered on a daily basis could increase the insight on environmental barriers that the MWU population must deal with. All these variables and factors are suspected to have a certain relationship with shoulder load and shoulder problems. Quantification of these WCMMs in real-life settings for longer durations and their associations with shoulder problems should be examined.

One area of interest for future research will be the development of an accurate and easy to use device or app for physical activity screening, health tracking for individuals or meeting activity goals in discussion and agreement with clinicians and therapists.

## 5. Conclusions

Summarizing the above, most WCMMs can be reliably derived during real-life conditions from one or two IMUs on a wheelchair. The distance covered, linear velocity, number of pushes and number and magnitude of turns can be reliably derived from one IMU on the wheelchair wheel. With an additional IMU on the frame, the inclination of the wheelchair can be reliably obtained. This opens the way to further examine real-life behavior in daily manual wheelchair ambulation.

## Figures and Tables

**Figure 1 sensors-23-07174-f001:**
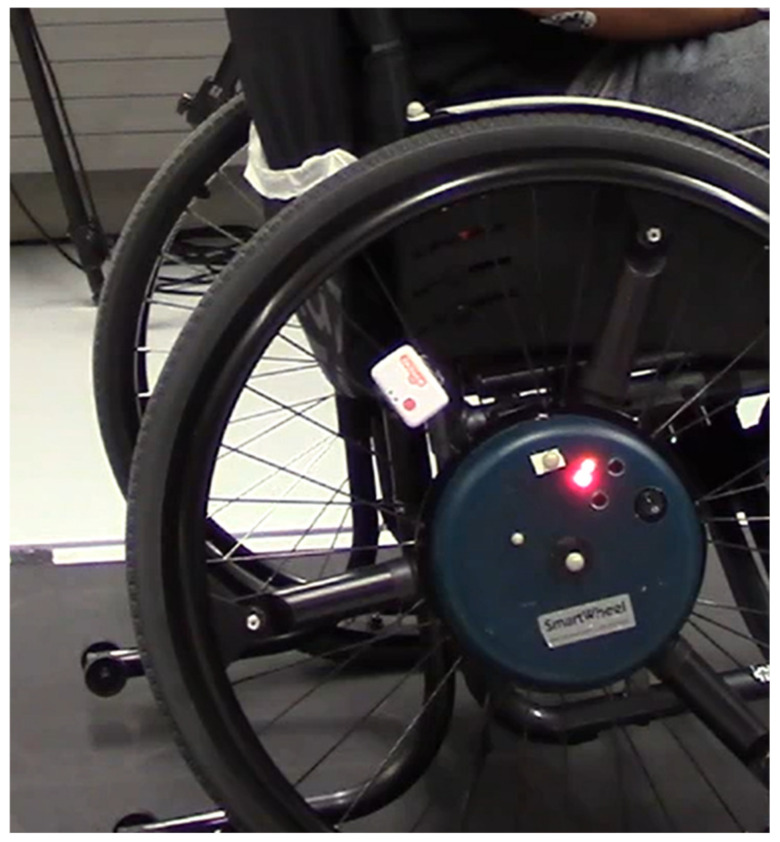
Equipment used. A standard active wheelchair with a Smartwheel on the right side, a Shimmer IMU attached to the wheel in such a way it did not hinder propulsion and another IMU attached to one of the horizontal bars of the wheelchair frame (not visible here).

**Figure 2 sensors-23-07174-f002:**
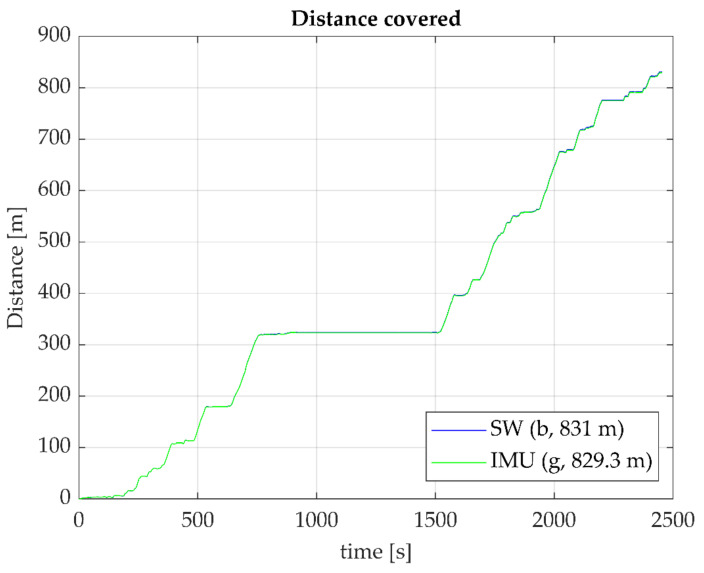
Distance covered as derived from Smartwheel data (blue) and from an IMU mounted on the wheel (green). Typical example from one participant. The blue line is hardly visible as the lines overlap, indicating equal performance over the complete trajectory.

**Figure 3 sensors-23-07174-f003:**
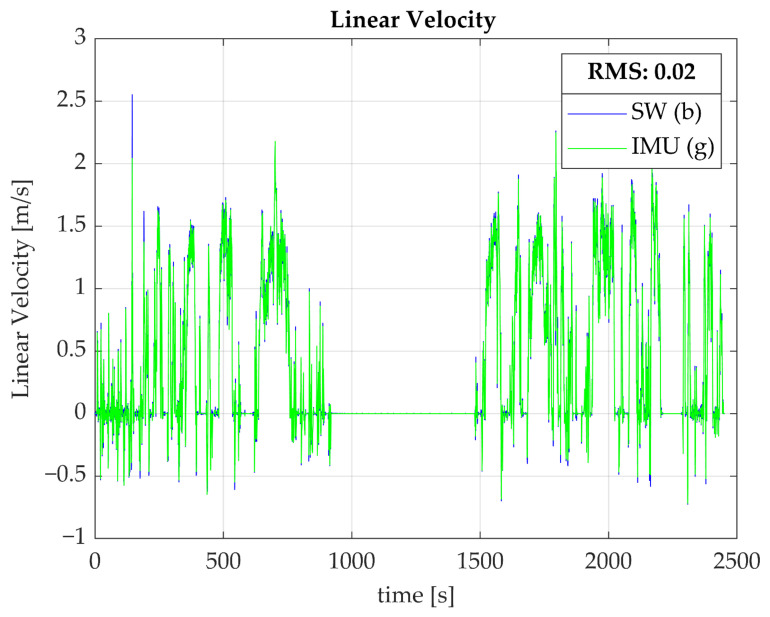
Linear velocity as derived from Smartwheel data (blue) and from an IMU mounted on the wheel (green). The graph represents a typical example from one participant.

**Figure 4 sensors-23-07174-f004:**
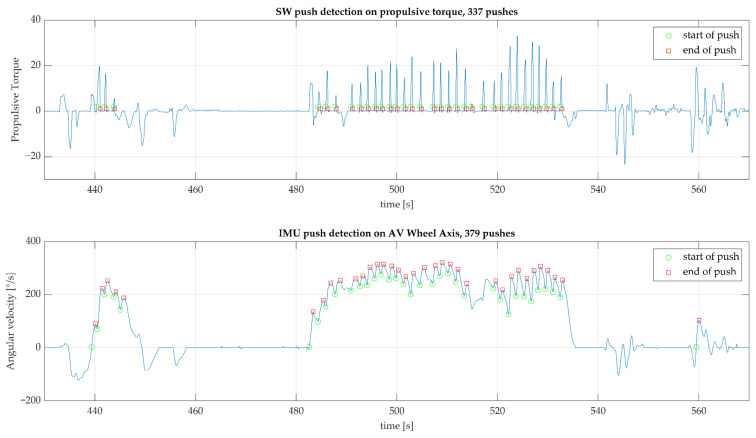
Typical example of the push detection for one participant using about two minutes of data. The top graph depicts the pushes detected from propulsive torque as measured with a Smartwheel; the lower graph depicts the results from push detection on angular velocity from an IMU attached to the wheelchair wheel. The number of pushes indicated in the titles of the graphs are for the full duration of the measurement, in this case around 40 min. A green circle indicates the start and a red circle the end of a push.

**Figure 5 sensors-23-07174-f005:**
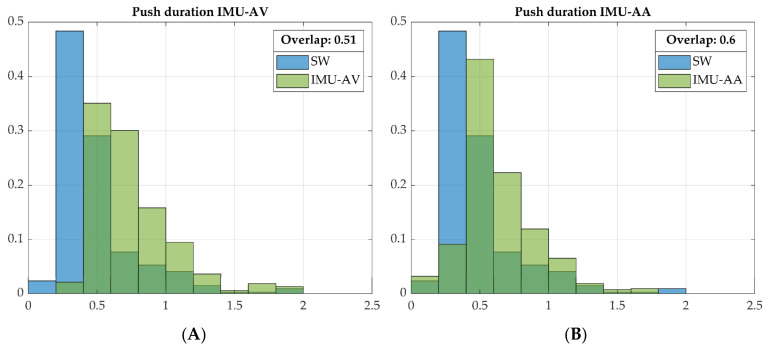
The normalized histograms show the overlap of push duration derived from the Smartwheel (SW, blue) and (**A**) IMU angular velocity (green) or (**B**) IMU angular acceleration (green) for one participant. Dark green areas visualize the overlap of the blue and green bars.

**Figure 6 sensors-23-07174-f006:**
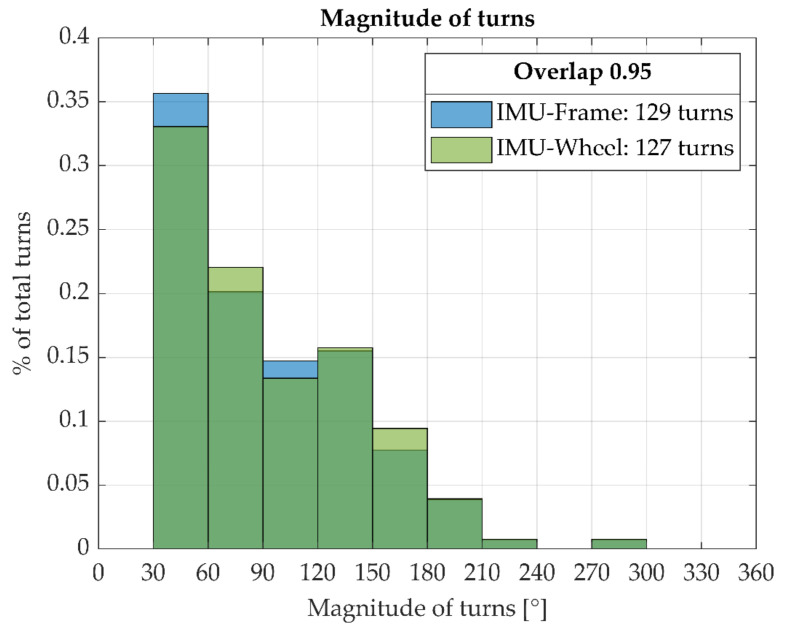
Normalized histograms for the magnitude of turns as derived from the data from the frame IMU (blue) and the wheel IMU (green) according to Equation (3). The percentage overlap (visualized in dark green) is calculated according to Equation (4); it is 95% for this specific participant.

**Figure 7 sensors-23-07174-f007:**
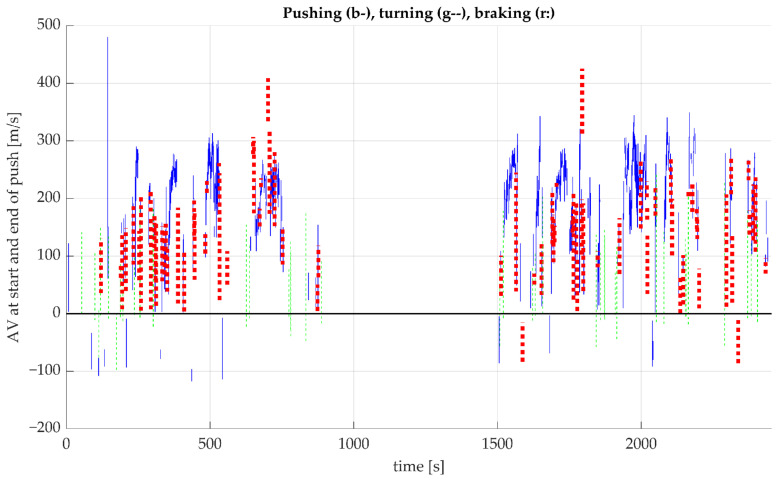
The graph depicts the different push styles that were observed in an extended analysis, indicating the need for clear definitions of pushes and braking actions. AV is angular velocity, which is positive when driving forward and negative when driving backward. The blue vertical lines indicate propulsive torque when driving forward (313 pushes), the dashed green lines indicate propulsive torque when driving backward and then forward during a push (26 pushes, making a turn on the spot, then driving forward) or when driving backward (red dotted lines, 2 pushes, in fact braking when driving backward); blue lines at negative angular velocity indicate negative torque when driving backward (11, propelling backward), the dashed green lines indicate negative torque when driving forward and backward during a push (12 times), the red dotted lines indicate negative torque while driving forward (75 times, in fact braking when driving forward).

**Table 1 sensors-23-07174-t001:** WCMM variables, averaged across participants, with error measures and ICC where applicable (two-way mixed, single measures, absolute agreement), across participants.

Variable	Reference Value	IMUs Value	Error	Error Measure ^1^	ICC	*p*-Value
Distance covered (m)	647.6	646.1	−0.2%	MAND	1.000	<0.001
Linear velocity (m/s)	-	-	0.02	RMS	-	
Number of pushes AV	429.6	428.6	4.1%	MAND	0.989	<0.001
Number of pushes AA	429.6	510.9	19.1%	MAND	0.869	<0.001
Push duration AV (s)	0.41	0.65	59.5%	MAND	0.030	0.159
Push duration AA (s)	0.41	0.54	33.1%	MAND	0.129	0.048
Cumulated WC turn (figure-8) (°)	1604	1600	0.65%	MAND	1.000	<0.001
Number of WC turns	86	88	3.4%	MAND	0.977	<0.001
Magnitude of turns		95.9%		% overlap	-	
Inclination long slopes ^2^	1.6, 1.8, 2.2	1.65, 1.9, 1.8	0.3	MAND	0.975 ^3^	0.000

AV = derived from angular velocity, AA = derived from angular acceleration, WC = wheelchair. ^1^ Error measures. MAND = mean absolute normalized difference, absolute difference normalized to reference value per participant; RMS = root mean square difference of the AV signals; % overlap = overlap of normalized histograms of wheelchair turns from frame and wheel IMU data. ^2^ Slope up and down; positive and negative values were compared. ^3^ ICC on averaged slope measures.

## Data Availability

The data presented in this study are available on reasonable request from the corresponding author.

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
