# Peer review of "Real-Life Wheelchair Mobility Metrics from IMUs"

_sensors, 2023, doi:10.3390/s23167174_

Round 1

Reviewer 1 Report

Thanks for giving me the opportunity to review this manuscript. I truly think this topic is important as can improve wheelchair users life from IMU´s use screening, as there is still a lot to do to accurately assess real life activity using the own wheelchair. We also realize the diversity of methodologies authors used to study such an event or activity (pushing / wheelchair propulsion) to maintain adequate ecological validity. However, in my opinion, some issues should be tackled and changes have to be made in order to consider it for final publication. Here are my suggestions:

We strongly suggest authors to also use the ICF model (WHO, 2001) in the background / discussion of their study, as it will help a lot in order to frame the benefits / outcome of the study using the main tool to share health issues. Please see for instance:

-          Gaspar, R., Padula, N., Freitas, T. B., de Oliveira, J. P., & Torriani-Pasin, C. (n.d. ). Physical Exercise for Individuals With Spinal Cord Injury: Systematic Review Based on the International Classification of Functioning, Disability, and Health, Journal of Sport Rehabilitation, 28(5), 505-516. https://doi.org/10.1123/jsr.2017-0185

-          Van Der Woude, L. H., Janssen, T. W., & Veeger, D. (2005). 3rd International Congress ‘Restoration of (wheeled) mobility in SCI rehabilitation: State of the art III’: its background. Technology and Disability, 17(2), 55-61.

Line 93. Data about male participants is missed, please include.

Line 138. “IMU mounted on the wheel could not be accurately aligned with the wheel axis. Therefore, a period of 10 seconds of straightforward wheelchair ambulation was selected from the recorded data (…)”. Was this procedure performed for every participant independently? Were those 10 seconds randomly selected from each participant records, or were recorded in the preprocessing protocol? We suggest authors to better clarify this information to allow a clear explanations and possible future study replications.

Line 180-190. Calculations of the number of pushes.

Line 191-193. Can you justify (even on previous research) why those criteria (30º and more than two seconds) to define a push?  

Line 230-231. “Only turns 230 larger than 30° were included”. Can you explain why?

Line 258 -260. “This variation in performance is representative for the target population of wheelchair users when including persons with paraplegia and tetraplegia”. In my opinion, this statement is not correct. We cannot infer from able-bodied just trained for the purpose of the study with persons with SCI wheelchair users in their daily living. This must be clarified and justified, as no trunk control has been assessed in this study, neither experience in wheelchair daily living nor wheelchair user interface mechanical features (centre of gravity height, for instance?). Many variables were not controlled so that statement cannot be assumed.

Line 349 – 350. Push duration present assessed differently with both methodologies used, and authors argue that the reason can the trunk involvement. Please see previous comment.

Future directions – conclusions. In our opinion, second part of the conclusions are in fact future research orientations. We suggest authors to rewrite both sections with a clear distinction between them. Also, an area of interest for future research must be the development of accurate and easy to use device for physical activity screening (to meet SCI population standards). Applications also to manual wheelchair sports assessment should be included in the future lines, as it has been cited sometimes along the text.

We do hope our suggestions will help authors to improve the manuscript.

Reviewer 2 Report

In this article, authors developed a methodology to measure wheelchair mobility metrics (WCMM) using inertial measurement units (IMUs) in real-life settings.  IMU-derived WCMM were validated against established methods and showed reliable estimation with low error and high Intra Class Correlation (ICC). However, push duration measured by IMUs differed from Smartwheel estimates due to the inclusion of thorax and upper extremity movements. It focus the daily wheelchair ambulation poses a risk factor for shoulder problems in manual wheelchair users with a reliable and unobtrusive methodology to better help to quantify the long-term effect of shoulder load. This information can contribute to the development of preventive measures, interventions, and therapies to mitigate shoulder problems in wheelchair users. Furthermore, the WCMM derived from IMU data have the potential for broader applications in individual health tracking and therapy settings beyond shoulder-related issues. However, there are a few areas that require clarification and improvement before publication.
1: Please provide a detailed description of the selection criteria for the 10 able-bodied participants who underwent the 40-minute wheelchair course. Are these participants representative of the manual wheelchair user population?

2:Clarify the process of validating the IMU-derived WCMM against Smartwheel and video analysis. Are there any limitations or biases in these reference methods?

3:Expand on the discussion of the other WCMM that were found to be reliably estimated from IMU data. How do these metrics contribute to understanding real-life behavior in wheelchair ambulation and their relevance to shoulder loading?

4:Review the manuscript for clarity and coherence. Some sentences are convoluted and could benefit from restructuring. Ensure that the research objectives, methodology, results, and implications are presented in a logical and organized manner.

5: It is advised to add pictures to illustrate the location of the sensor unit on the wheelchair.

The quality of English in the provided summary and peer review comments is strong. The sentences are well-structured, concise, and effectively convey the intended meaning. The language used is appropriate for a scientific context, demonstrating a good command of technical terminology. The writing is clear and coherent, allowing for easy understanding of the research topic, methodology, and findings. 

Reviewer 3 Report

This manuscript presents a methodology to obtain wheelchair mobility metrics (WCMM) from IMU's located on the wheelchair frame and wheel in real-life settings. These metrics include distance covered by the wheelchair, linear velocity of the wheelchair, number and magnitude of turns and inclination of the wheelchair, and number and duration of pushes. However, this manuscript must be improved based on the following issues:

1.-The authors used IMU sensors. The title should include wearable sensors. Although, wearable sensors add more sensor types. The title must be more specific by considering the used sensors.

2.-The introduction section should consider the main advantages and limitations of the proposed method for WCMM using IMU sensors in comparison with other methods reported in the literature. 

3.-This manuscript should add a comparative table with the main parameters, advantages, and drawbacks of the proposed method with respect to others reported in the literature.

4.- The second section should include schematic views or figures that help to describe the proposed methodology. For instance, the position of the sensors on the wheelchair and the inclination of the wheelchair. 

5.-The numeration of the figures must be corrected.

6.- The description of the proposed methodology must consider more detailed information on the main parameters of the sensors. 

7.- The authors must enhance the discussions of the main results.

8.-What are the statistical data of the main results?

9.-What are the challenges of the proposed methodology?

10.- What are the future research works?

11.- The quality and resolution of figures 1-6 must be improved.

12.-The conclusions must be modified by considering the above comments.

13.-The format of references is different from that used in MDPI. 

The English grammar can be enhanced.

Round 2

Reviewer 3 Report

The authors have addressed all the reviewer's comments.

The English grammar can be enhanced.